# Study on the Vibration Isolation Performance of Composite Subgrade Structure in Seasonal Frozen Regions

**Leilei Han** , **Haibin Wei and Fuyu Wang** *

School of transportation, Jilin University, Changchun 130022, China; hanll18@mails.jlu.edu.cn (L.H.);
weihb@jlu.edu.cn (H.W.)

* Correspondence: wfy@jlu.edu.cn; Tel.: +86-151-431-73491



**Featured Application: There is a great potential application of the research findings in the respect of highway subgrade in seasonal frozen regions.**

**Abstract:** Silty clay modified by fly ash and crumb rubber is a kind of sustainable subgrade filler that has good freeze–thaw resistance stability, but weak vibration isolation performance. The objective of this study was to improve the vibration isolation of the modified soil and investigate the vibration isolation effect of the composite subgrade structure of extruded polystyrene (XPS) plates and the modified soil by the indoor impact test. First, the vibration isolation performance of silty clay, modified soil, and composite subgrade structure was respectively evaluated. Second, the effect of the XPS plate's thickness and vibration intensity on the vibration performance of the composite subgrade structure were evaluated. Third, the vibration isolation performance of the test groups under the condition of freeze–thaw cycles was assessed. The results show that the vibration isolation performance of the subgrade can be effectively improved by setting XPS plates. The composite subgrade structure has a certain vibration isolation effect, especially in the vertical direction. Considering the vibration isolation performance and costs, 5 cm was the optimum XPS plate thickness. The composite subgrade structure showed a great vibration isolation performance under the condition of freeze–thaw cycles, so it is suitable for application in road subgrade in seasonal frozen regions.

**Keywords:** XPS plates; composite subgrade structure; vibration isolation; impact load test; freeze–thaw cycles

## 1. Introduction

The problem of environmental vibration caused by traffic load has attracted more and more attention with the rapid development of road traffic in recent years. Road traffic accounts for 11.3% of the total complaint rate of environmental vibration pollution and ranks third behind construction and factories, according to relevant statistics [1].

Vibration caused by traffic load may result in the strength decrease and settlement increase of roadbed and foundation. The long-term vibration can cause serious damage to nearby buildings, threaten the comfortability of people's lives, and affect the normal use of precision instruments [2,3], which must be solved urgently. To solve the vibration pollution caused by trains, Costa et al. [4] presented a model to understand the dynamic behavior of ballasted tracks with mats. It was found that the vibration amplitude could be reduced by placing the mat beneath the subballast. Bajcar et al. [5] conducted in-site experiments to measure the vibration levels of buried operating natural gas pipelines under the moving traffic. They focused on the assessment of the impact of the traffic-induced vibration

and the increased risk on individuals that were affected by such underground natural gas pipelines at road crossings.

Based on soil stabilization technology, a novel subgrade material that is silty clay modified by fly ash and crumb rubber was proposed by our research group. It has been proven that the modified soil possesses greater performance in conventional physical properties of strength, deformation, and stability, especially after freeze–thaw cycles [6–8]. This modified soil is also regarded as a sustainable subgrade material because it disposes and recycles the large amount of industry waste fly ash and rubber products. The great resistance to freeze–thaw cycles contributes to the sustainable performance of the subgrade in seasonally frozen regions. However, many sources show that the anti-vibration performance of soils modified by fly ash is poor, as the liquefaction resistance decreases after multiple vibrations [9,10]. The research on the anti-vibration of silty clay modified by fly ash and crumb rubber is scarce and worth studying.

The influence of vibration on the sustainable performance of buildings can be effectively mitigated by the design and development of more efficient solutions. In road engineering, an isolation layer is effectively adopted against vibrations. Expanded polystyrene foam sheet (EPS) is the most commonly used isolation material, which has the characteristics of ultra-light weight, good compressive property, high strength, good durability, and thermal insulation [11,12]. Xiang et al. [13] proved the significant vibration isolation effect of EPS on soils by the indoor model test. It was found that the closer the EPS board and the vibration generator, the better the vibration isolation effect. Murillo et al. [14] studied the viability of EPS as the vibration isolation layer. The effect of barrier depth, thickness, and distance from the vibration source on the vibration isolation of EPS was investigated.

XPS is a rigid foam board made of polystyrene resin, other raw materials and polymers, heated, mixed, and injected with the catalyst at the same time, and then extruded and molded. Compared with EPS, XPS (extruded polystyrene) foam sheets have weaker hygroscopicity, lower thermal conductivity, higher compressive strength, and more environmentally friendly characteristics [15,16]. Furthermore, it has also been used as the functional layer (i.e., thermal insulation, vibration isolation, and so on) beneath the building's foundation. Kilar et al. [17] performed laboratory tests to evaluate the performance of XPS board in earthquake engineering and also revealed the failure mechanism of the building structure founded on XPS board subjected to earthquake loading. Ertugrul et al. [18] found that the presence of foam behind the flexible retaining wall could effectively result in a reduction of dynamic pressure and displacement. Mao et al. [19] summarized the achievements of protecting the Qinghai-Tibet Plateau permafrost by XPS board and series of valuable guidance regarding design, construction, and quality control measures were given.

It is more valuable and promising to replace EPS with XPS board. However, the investigation into the application of EPS in practical engineering is much more advanced than the corresponding research performed on XPS, especially in road engineering. In this study, XPS plates were adopted to improve the vibration isolation performance of silty clay modified by fly ash and crumb rubber. The composite subgrade structure of XPS plates and the modified soils was called the composite subgrade structure. The objective of this paper was to: (1) evaluate the vibration isolation effect of silty clay, the modified soil, and the composite subgrade structure of the modified soil with XPS plates; (2) determine the optimum thickness of XPS plates in the composite subgrade structure; and (3) investigate the effect of vibration intensity and freeze–thaw cycles on the vibration isolation of the composite subgrade structure.

## 2. Materials and Methods

### 2.1. Materials

#### 2.1.1. Raw Materials

The soil used in this test was silty and its physical properties are listed in Table 1. Fly ash was silica alumina fly ash, which was provided by the Power Plant of Changchun. The content of $SiO_2 + Al_2O_3$ was 78.13~88.64%, CaO was 4.12~7.02%, $SO_3$ was 0.1~0.72%, and loss of ignition was 1.22~5.26%.

**Table 1.** Typical physical properties of silty clay.

| Liquid Limit | Plastic Limit | Plasticity Index | Optimum Water Content | Maximum Dry Density |
|---|---|---|---|---|
| 34% | 22.40% | 11.60% | 12.1% | 1.92 g/cm$^3$ |

The rubber particles were collected from the Changchun Rubber Products Factory with grains size between 1 mm and 1.5 mm. These were the recycled products of waste rubber tires.

XPS plates were made by Jilin Fuquan New Thermal Insulation Material Co. Ltd. The products had a flat surface, no inclusions, and uniform color. Its compressive strength was 256 kPa.

### 2.1.2. Preparation of the Modified Soil

First, the raw silty clay and fly ash were dried, crushed, and sieved through the sieve of 2 mm. Second, they were mixed at the dry mass ratio of 1:2. Then, rubber particles, which accounted for 2% of the total mass, were added. Finally, all ingredients were mixed evenly after the calculated amount of water was added. According to the previous research of [6–8], the modified soil in this mixture ratio has fairly good anti-freeze–thaw stability as a green and sustainable road subgrade material with good performance for seasonal frozen regions. Before testing, the mixture was placed in the humidor for 24 h. The physical indicators of the modified soil are listed in Table 2.

**Table 2.** Physical indicators of the modified soil.

| Liquid Limit | Plastic Limit | Plasticity Index | Optimum Water Content | Maximum Dry Density |
|---|---|---|---|---|
| 36.0% | 25.1% | 10.9% | 19.2% | 1.61 g/cm$^3$ |

### 2.2. Impact Load Test

### 2.2.1. Testing Apparatus

The test model box (dimension of 0.3 m × 0.3 m × 0.5 m) was made of iron plates (thickness of 3 mm). The boundary reflection seriously affects the test results, especially in the case of a small size and rigid boundary. To reduce this boundary reflection, XPS plates (thickness of 3 cm) were placed at the bottom and surrounding the model box. The small rigidity and high porosity of XPS could effectively reduce the boundary reflection.

The piezoelectric acceleration sensor DH311E and the acquisition instrument DH5922 used in this study was made by Donghua Co. Ltd. (Jingjiang, China). To avoid being affected by moisture and soil when obtaining data, the acceleration sensor should be wrapped with waterproof plastic bags.

### 2.2.2. Impact Load

A drop hammer was used to generate the vibration when applying the impact load. The impact load applying device was composed of a drop hammer, a connecting bar, and a round bottom plate (diameter of 9 cm, thickness of 1 cm).

Impact loads with different magnitudes can be achieved by adjusting the drop height of the hammer. The impact force can be calculated according to the law of conservation of momentum. It is considered that the plate provides the hammer with a counter impulse with the equal value but opposite direction of the impact impulse. An acceleration sensor was placed on the bottom plate to calculate the counter impulse. Figure 1 shows the diagram of collected acceleration versus time. The product value of the plate mass (m) and the area enclosed by the acceleration curves with time (a*t) is equal to the value of counter impulse (F*t).

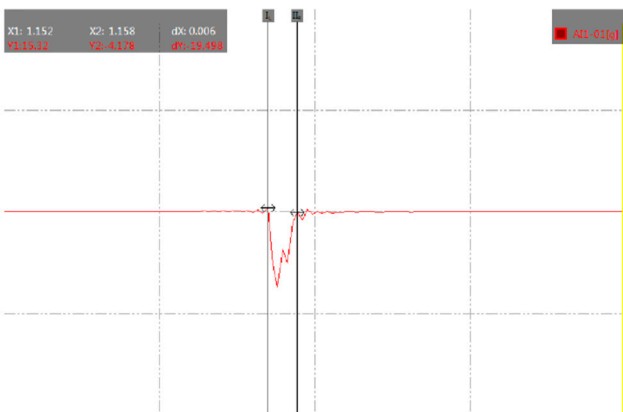

**Figure 1.** Diagram of the collected acceleration versus time for the impact load.

This counter impulse can be simulated by the triangular pulse shape [20,21], which is presented in Figure 2. Just as shown, $F_1$ is the maximum point of the counterforce and $T_1$ is the interaction time. $F_1$ is regarded as the load force in the analysis. In this study, the action time of the impact load was about 0.006 s. Based on the collected data of the load test, the impact load force and the corresponding stress of the drop hammer falling from different heights were calculated, which are listed in Table 3.

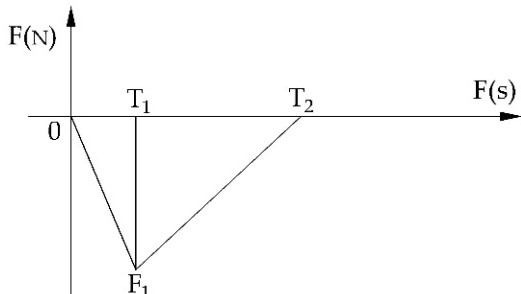

**Figure 2.** Equivalent triangular waveform of the impact load.

**Table 3.** The impact force and stress corresponding to the falling height.

|  | Drop Height (cm) | | | | |
|---|---|---|---|---|---|
|  | **5** | **10** | **15** | **20** | **25** |
| Impact force (N) | 100.78 | 142.52 | 174.55 | 201.55 | 225.34 |
| Impact stress (Pa) | 15845.91 | 22408.81 | 27444.97 | 31690.25 | 35430.82 |

2.2.3. Preparation of the Test Soils and Sensors in the Mold Box

According to the calculated compaction degree, the test soils were poured into the mold (five layers, the thickness of each layer was 10 cm) and compacted by a hammer. In the process of compaction, acceleration sensors were buried at the designed depth. Attention should be given to the accuracy of the location of the sensors. Furthermore, to investigate the application viability of this composite subgrade structure in seasonally frozen areas, the vibration isolation test under freeze–thaw cycles was conducted. A temperature sensor was also buried in the mold to monitor the freeze–thaw cycle process. According to the previous test data before this test, the standard of one freeze–thaw cycle was set as freezing for 24 h at a temperature of −15 °C and thawing for 24 h at a temperature of 15 °C. Then, the test was carried out at room temperature, and the temperature collected by the temperature sensor was between 15 °C and 16 °C at the end of the test.

### 2.2.4. Test Procedures

Three test groups, the silty clay, the modified soil, and the composite subgrade structure, respectively, were designed to study the vibration isolation effect. The three test groups were named as G (group) 1, G2, and G3, respectively, and G1 and G3 are presented in Figure 3. For G1, the mold box was completely filled with silty clay. As for the other two groups, the upper 30 cm of the soils were replaced by the modified soil and the composite subgrade structure, respectively. The sensor was buried at the depth of 30 cm, which was exactly the bottom of the modified soils. In G3, different thicknesses (3 cm, 5 cm, and 9 cm) of the XPS plates were used to determine the optimum thickness of the XPS plates. When testing, a slab of bitumen mixture (the size of 5 cm × 20 cm × 20 cm) was placed at the top of the test groups. The impact applying device was put on the surface of the asphalt mixture. The drop height of the hammer was set as 5 cm, 10 cm, 15 cm, 20 cm, and 25 cm to simulate the vibration waves of different intensities. As the drop hammer fell down, the vibration waves generated and propagated through the different layers. Accelerations of the fierce vibration could be collected by the sensor. The vibration isolation effect of the three test groups was assessed by a series of impact load tests.

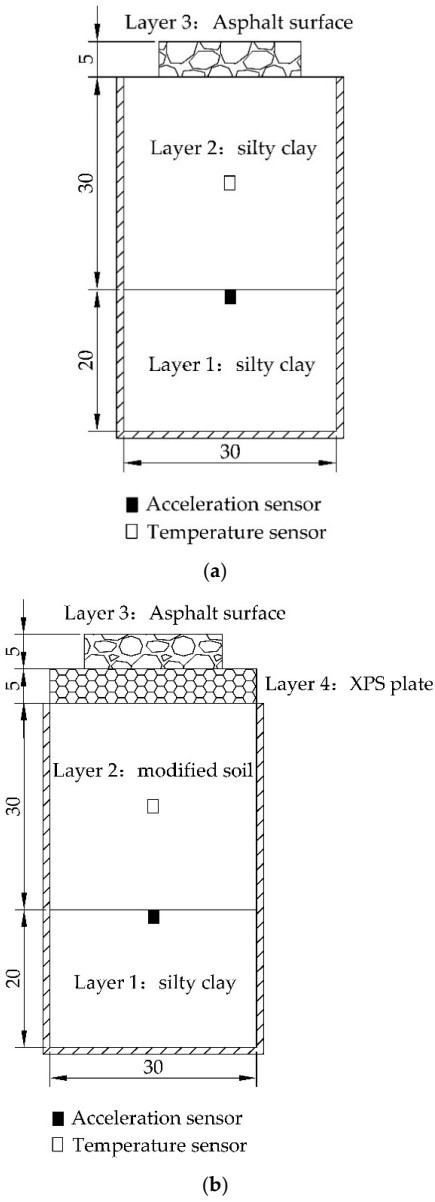

**Figure 3.** Diagram of sensors and filling materials of the two test groups. (**a**) G1; (**b**) G3.

## 3. Results and Discussion

In this study, ten repeated operations for each test condition were designed. The transverse (X-direction and Y-direction) and the vertical (Z-direction) acceleration vibration waves are collected by the sensor, which is illustrated in Figure 4 as an example. In Figure 4, the horizontal axis represents time (s) and the vertical axis represents acceleration (m/s²). The peak acceleration value of each wave is recorded, and the average value of ten acceleration peaks is calculated as the representative value to conduct the analysis of the vibration isolation effect. The results of the indoor impact test under different test conditions (different thickness of XPS plates and different falling heights of the drop hammer) are listed in Table 4. It should be mentioned here that the direction of the traffic load was not included in this study, so the acceleration values were supposed to be no different in the X- and Y-direction. However, there was a certain difference between the acceleration values of Table 4, which may have resulted from the fact that the load was applied and the sensor was not completely in the center of the X–Y plane.

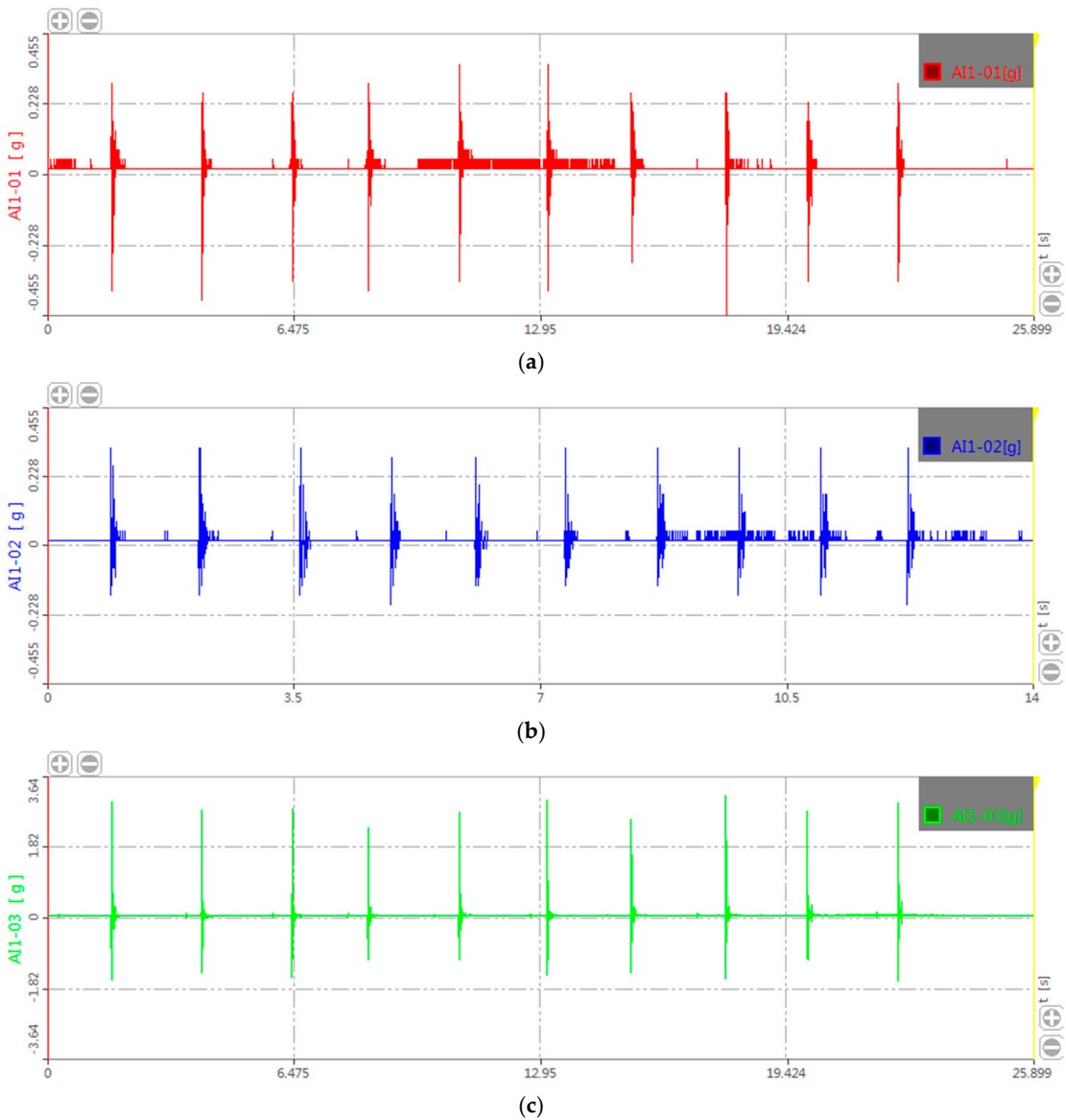

**Figure 4.** Measured acceleration wave curves. (**a**) Acceleration of the X-direction; (**b**) Acceleration of the Y-direction; (**c**) Acceleration of the Z-direction.

**Table 4.** Peak value of acceleration under different test conditions. (Unit: g).

| Height of the Fall | 5 cm | | | 10 cm | | | 15 cm | | |
|---|---|---|---|---|---|---|---|---|---|
| | X- | Y- | Z- | X- | Y- | Z- | X- | Y- | Z- |
| G1 | 0.1908 | 0.1606 | 0.9825 | 0.194 | 0.2238 | 1.3739 | 0.233 | 0.2815 | 1.621 |
| G2 | 0.1818 | 0.272 | 1.2744 | 0.203 | 0.2873 | 1.7267 | 0.2815 | 0.2871 | 2.0253 |
| G3 (3 cm XPS) | 0.0848 | 0.1481 | 0.1565 | 0.1396 | 0.2422 | 0.2819 | 0.1879 | 0.321 | 0.4096 |
| G3 (5 cm XPS) | 0.0819 | 0.094 | 0.0998 | 0.1213 | 0.1542 | 0.1737 | 0.1427 | 0.251 | 0.2589 |
| G3 (9 cm XPS) | 0.076 | 0.082 | 0.0767 | 0.088 | 0.1546 | 0.1027 | 0.1335 | 0.203 | 0.1253 |
| Height of the Fall | 20 cm | | | 25 cm | | | | | |
| | X- | Y- | Z- | X- | Y- | Z- | | | |
| G1 | 0.2875 | 0.2964 | 1.8718 | 0.2756 | 0.3539 | 2.1532 | | | |
| G2 | 0.336 | 0.3693 | 2.4321 | 0.3903 | 0.4507 | 2.7335 | | | |
| G3 (3 cm XPS) | 0.2242 | 0.354 | 0.4894 | 0.2905 | 0.369 | 0.5692 | | | |
| G3 (5 cm XPS) | 0.1907 | 0.2752 | 0.2962 | 0.212 | 0.2968 | 0.3584 | | | |
| G3 (9 cm XPS) | 0.164 | 0.257 | 0.1647 | 0.176 | 0.2814 | 0.205 | | | |

### 3.1. The Analysis on the Vibration Isolation Effect of Test Groups

To compare the vibration isolation effect of silty clay, the modified soil, and the composite subgrade structure, the relationship curves between the acceleration peak and falling heights of hammer for the three groups were drawn, which are depicted in Figure 5.

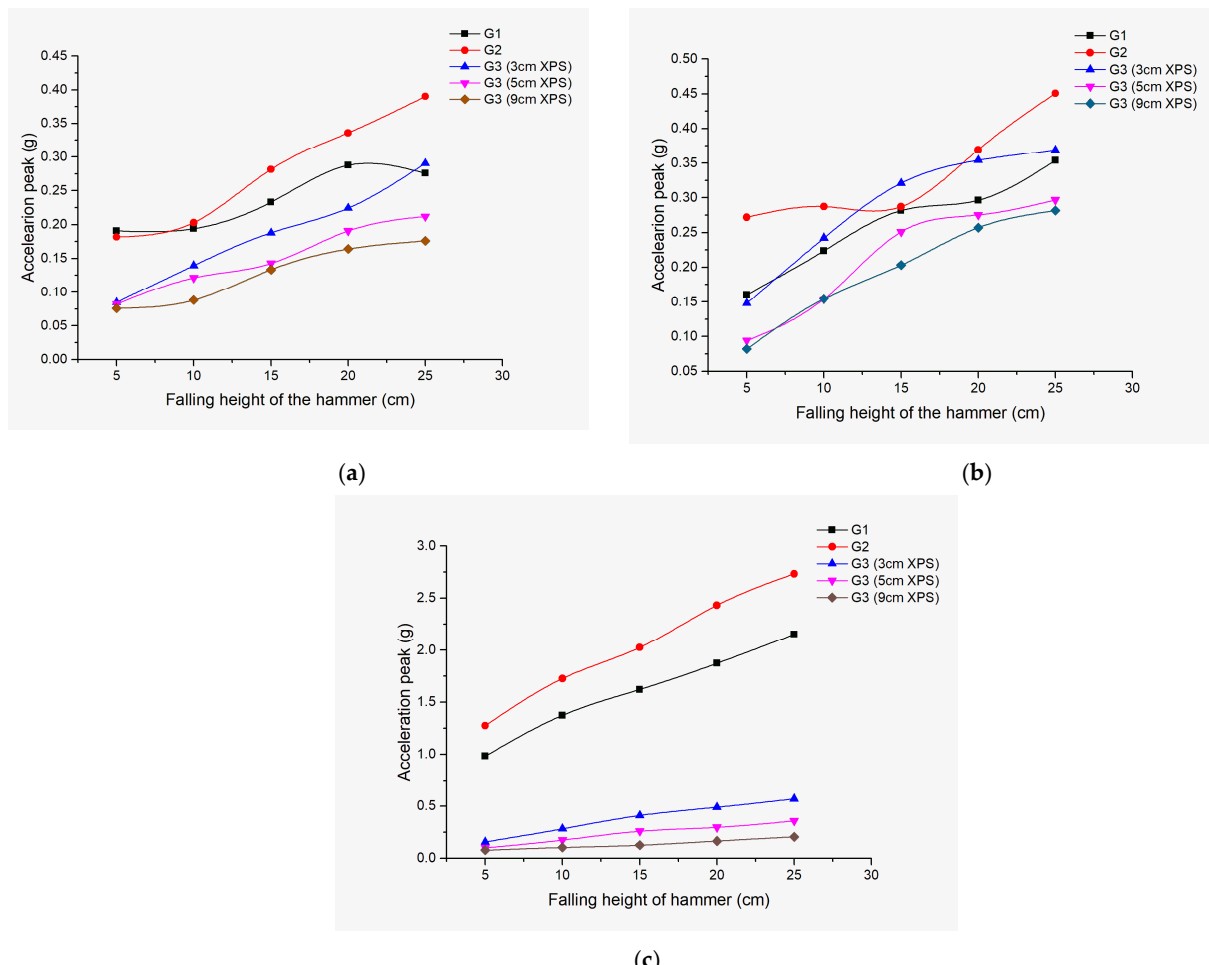

(a)

(b)

(c)

**Figure 5.** Acceleration peak values of the three test groups. (**a**) Acceleration peak in the X-direction; (**b**) Acceleration peak in the Y-direction; (**c**) Acceleration peak in the Z-direction.

From Figure 5, the acceleration peak values in the three main axes (X, Y, and Z-direction) for G1, G2, and G3 embedding different thicknesses of XPS plates rose as the falling heights increased. Comparing the data of G1 to G2, the acceleration peak values of G1 were lower than that of G2, especially in the Z-direction. This means that the vibration isolation performance of the modified soil was worse than that of silty clay. The reason for this phenomenon is that fly ash reduces the damping of improved soil [22], while rubber particles increase the damping of modified soil. The strengthening effect of 2% rubber particles is very limited. The 32% content of fly ash dominates the change of damping of modified soil, which leads to the phenomenon that the damping of modified soil is less than that of silty clay and the vibration amplitude of modified soil is larger than that of silty clay. Thus, it is necessary to improve the vibration isolation performance of the modified soil subgrade.

Just as shown in Figure 5, the acceleration peak values of G3 were basically lower than that of G1, except for the accelerations of G3 with XPS plates of 3 cm in the X-direction and Y-direction. The results prove the viability of using XPS plates to improve the vibration isolation effect of the modified soil, especially using the thicker XPS plates. This phenomenon is attributed to the composition and structure of foam boards. Both EPS and XPS plates are made of polystyrene foam. The large amount of pores in the foam plate are beneficial to absorb the vibration and fluctuation [16,23]. The perfect closed-cell honeycomb structure makes the XPS plates have excellent stability, which is an important reason as to why it can be used as a light subgrade material.

Furthermore, it was noted that the acceleration peak values in the Z-direction were far higher than that of the X-direction and Y-direction. For road engineering, the vibration generated by traffic loads would propagate downward in the pavement structure. The vibration isolation effect in the Z-direction was of most concern because the vibration amplitude in the Z-direction was much larger than that in the other two directions, which had the greatest influence on the structure. The vertical vibration is considered as a vital parameter for measurement and construction control.

### 3.2. Influencing Factors of Vibration Isolation Effect for the Composite Subgrade Structure

The previous section has proven the great vibration isolation of the composite subgrade structure. In order to investigate the vibration isolation effect of the composite subgrade structure with different thicknesses of XPS plates or vibration intensity (corresponds to different falling heights of the hammer), the following analysis was conducted.

In this study, the isolation coefficient was adopted to evaluate the vibration isolation effect of the composite subgrade structure, which is defined as:

$$C = A_i/A_0 \tag{1}$$

where C is the isolation coefficient; $A_i$ is the acceleration peak value of the composite subgrade structure under different test conditions; and $A_0$ is the acceleration peak value without setting the composite subgrade structure (corresponds to the acceleration value of G1). When $C < 1$, it means that the vibration isolation effect is significant. The smaller the C value, the better the vibration isolation effect.

### 3.2.1. The Effect of XPS Plate Thickness on the Vibration Isolation

Thickened XPS plates can effectively improve the vibration isolation performance of the composite subgrade structure. However, it sometimes causes a negative impact of overall structure and increases the usage cost. Therefore, it is vital to determine the optimum XPS plate thicknesses of the composite subgrade structure. In this part, the thicknesses of the XPS plates were set as 3 cm, 5 cm, and 9 cm, respectively. The impact load was generated by hammer dropping from the height of 25 cm. The result is illustrated in Figure 6.

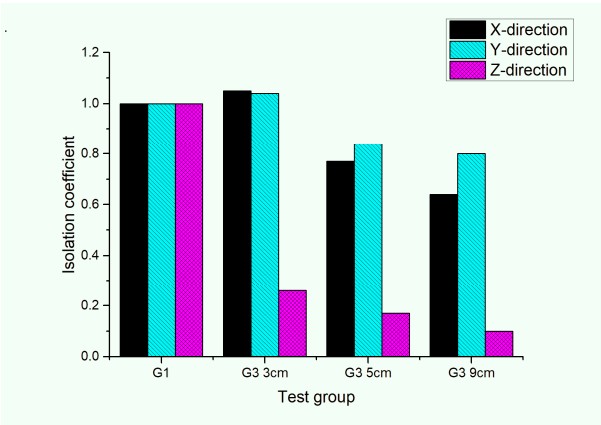

**Figure 6.** Isolation coefficient of the composite subgrade structure.

From Figure 6, the isolation coefficient of the composite subgrade structure decreased as the thickness of the XPS plates increased. For G3, which embeds XPS plates with the thickness of 9 cm, the isolation coefficient C in the three main axes were 0.64, 0.80, and 0.10, respectively. The vibration isolation effect in the Z-direction was more significant than that in the other two directions. It should be noted that the C value in the X-direction and Y-direction for G3 embedding with 3 cm XPS plates was greater than for G1. This means that the vibration isolation in transverse performs a negative effect when compared to G1. However, the vibration isolation effect in the Z-direction of the composite subgrade structure is the most concerning problem. The relationship between the isolation coefficient in the Z-direction and the thickness of XPS plates is fitted in the exponential form, which is demonstrated in Figure 7.

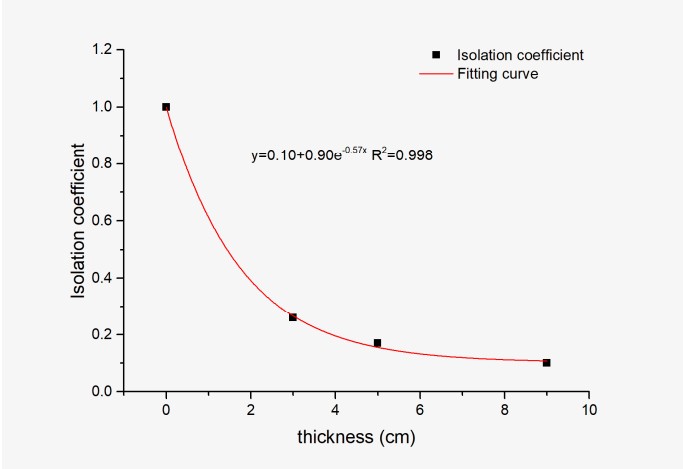

**Figure 7.** Fitting curve of the isolation coefficient in the Z-direction.

As shown in Figure 7, the XPS plates with thicknesses of 5 cm and 9 cm both possessed significant vibration isolation performance. The effect of the composite subgrade structure with 9 cm was slightly better. However, the enhancement in isolation performance was very limited when the thickness of the XPS plates increased from 5 cm to 9 cm. Furthermore, the cost of 9 cm XPS plates was much higher than that of 5 cm. Therefore, the XPS plate thickness of 5 cm was considered as the optimum thickness of the composite subgrade structure in these test groups. The isolation coefficient C of the composite subgrade structure embedding 5 cm XPS plates in Z-direction was 0.17, which accords with the normal vibration isolation standard [24].

### 3.2.2. The Effect of Vibration Intensity on the Vibration Isolation

In actual engineering, traffic loads with different weights always produce vibrations of different intensities. In this part, the vibration isolation performance of the composite subgrade structure subjected to different vibration intensities was investigated. The vibration of different intensities was achieved by a load hammer falling from different heights (5, 10, 15, 20, and 25 cm). The thickness of the XPS plates was 5 cm, which was determined in the previous section. The results are shown in Figure 8.

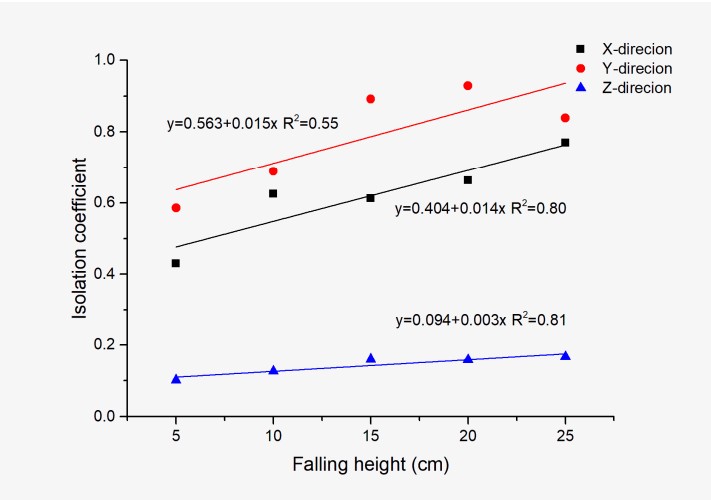

**Figure 8.** The isolation coefficient of the composite subgrade structure under different falling heights.

Figure 8 shows that the isolation coefficient of the composite subgrade structure increased as the falling height increased. To master the characteristics of the isolation coefficient with the change in vibration intensity, the test data were fitted in linear forms, which are demonstrated in Figure 8. From Figure 8, it can be seen that the correlation coefficients of the X-direction and Y-direction were stronger than that of the Z-direction. The slope value represents the decay rates of the isolation coefficient with the increase in vibration intensity. The slope of the X-direction, Y-direction, and Z-direction were 0.014, 0.015, and 0.03, respectively. The decay rate of the isolation coefficient in the Z-direction was the lowest, which means that the vibration isolation effect in the most concerned Z-direction has the smallest change with the increase in vibration intensity, showing good vibration isolation stability. Specifically, the vibration isolation coefficient increased by 62.7% (from 0.102 to 0.166) while the load intensity increased by 400% (from 5 cm to 25 cm). The vibration isolation effect of the X-direction and Y-direction varied greatly with the increase in the vibration intensity. However, the vibration amplitude of the X-direction and Y-direction was far less than that of the Z-direction when the XPS plates were not set. In addition, the vibration isolation coefficients were all less than 1 in this study, which means that the vibration intensity of the XPS plate structure was less than that of the unset structure, so the vibration influence in the X-direction and Y-direction is very limited. When the falling height of the hammer was 25 cm, the isolation coefficient in the Z-direction was 0.166, which was still lower than 0.2. Thus, it can be concluded that the composite subgrade structure possessed great vibration isolation performance in the Z-direction, even under great intensity vibration.

### 3.3. Vibration Isolation of the Composite Subgrade Structure after Freeze–Thaw Cycles

The main reason for road diseases in seasonally frozen areas is that the stability and durability of the subgrade material decreases after freeze–thaw cycles, which results in the increase of soil subsidence deformation and uneven sinking of the road. It is necessary to ensure the anti-freeze–thaw performance of the composite subgrade structure in seasonally frozen areas. The acceleration peak value of the composite subgrade structure embedding 5 cm XPS plates after 1, 2, 3, 4, and 5 freeze–thaw

cycles were measured. The vibration isolation analyses in the three main axes were conducted as well as the comparative analysis with G1 and G2. The results are demonstrated in Figure 9.

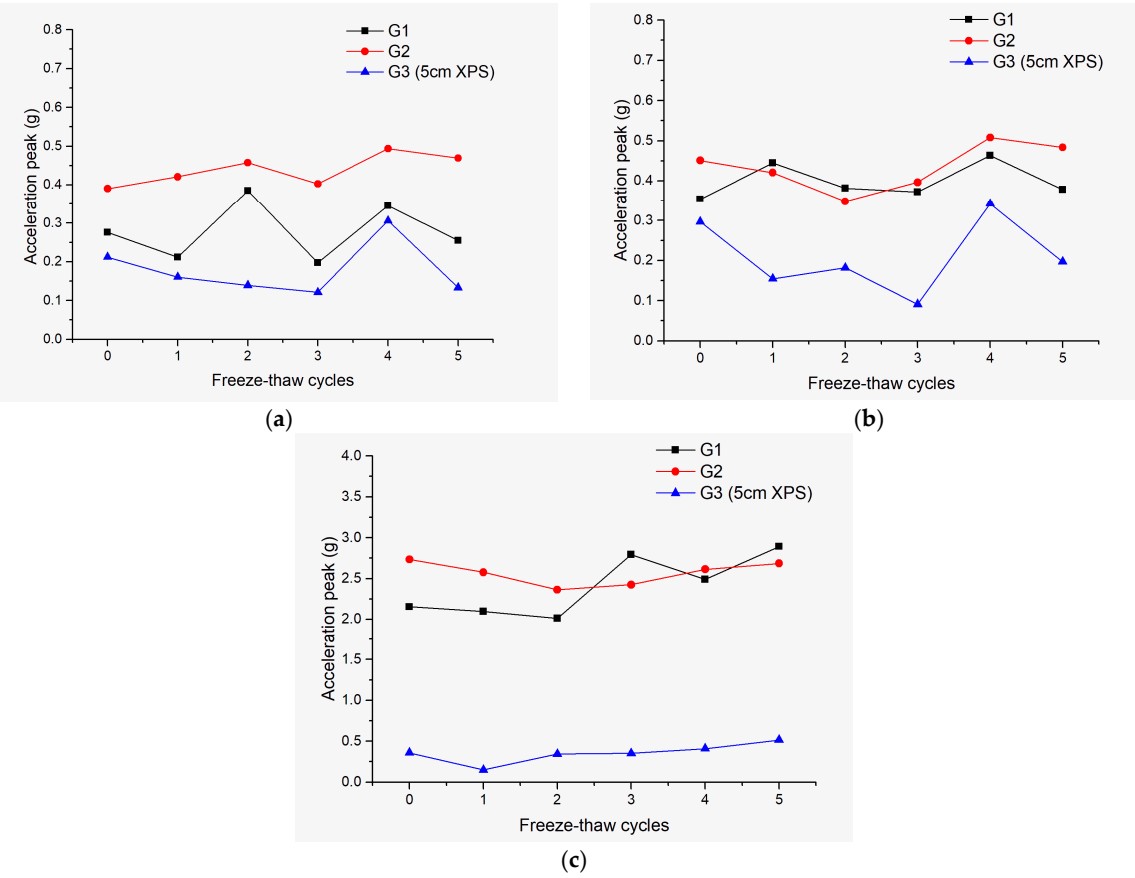

**Figure 9.** Acceleration peak of the three test groups after freeze–thaw cycles. (**a**) Acceleration peak in the X-direction; (**b**) Acceleration peak in the Y-direction; (**c**) Acceleration peak in the Z-direction.

From Figure 9, the acceleration peak values of the three test groups fluctuated with the increase in the number of freeze–thaw cycles, especially G1 (filled by silty clay). To better describe the fluctuation characteristic of the acceleration peak, statistical analysis was performed to calculate the mean, vibration amplitude range, and standard deviation of G1, G2, and G3. The results are listed in Table 5.

**Table 5.** The static results of the three test groups.

| Direction | Type | Mean (g) | Range (g) | Standard Deviation (g) |
|---|---|---|---|---|
| | G1 | 0.278 | 0.188 | 0.074 |
| X-direction | G2 | 0.439 | 0.103 | 0.041 |
| | G3 (5 cm XPS) | 0.179 | 0.185 | 0.070 |
| | G1 | 0.399 | 0.109 | 0.044 |
| Y-direction | G2 | 0.435 | 0.160 | 0.059 |
| | G3 (5 cm XPS) | 0.211 | 0.251 | 0.093 |
| | G1 | 2.406 | 0.878 | 0.376 |
| Z-direction | G2 | 2.566 | 0.370 | 0.135 |
| | G3 (5 cm XPS) | 0.355 | 0.364 | 0.139 |

Among the three test groups, the mean values of G2 in the three directions were all the largest, which means that the vibration isolation of the modified soil was poor and the reason has been explained in Section 3.1. However, the standard deviations and fluctuation ranges of G2 were small,

which represents that its stability of freeze–thaw resistance was great. Comparing the results of G1 to G2, after five freeze–thaw cycles, the acceleration peak in the Z-direction of G2 was close to that of G1. This means that the vibration isolation performance of G2 was similar to G1 after five freeze–thaw cycles. However, the dynamic response stability of freeze–thaw resistance for G2 was far better than that of G1, and indicates that the modified soil performed better than the silty clay when they underwent the freeze–thaw cycles, which is the reason why the modified soil is selected as a better subgrade filler in road engineering in seasonally frozen areas. The mean and range of G3 in the Z-direction was the smallest. The mean value decreased by 86.2% (2.211 g) when compared with G2. The range and standard deviation were basically consistent with G2, decreasing by 1.6% (0.006 g) and increasing by 2.9% (0.004 g), respectively. These data fully prove the high efficiency and stability of the vibration isolation effect of the composite subgrade structure.

From the above results, it can be concluded that the vibration isolation of the composite subgrade structure after several freeze–thaw cycles was still significant, especially in the Z-direction. The isolation coefficients of the composite subgrade structure after the freeze–thaw cycles are described in Figure 10. The isolation coefficients in the X-direction and Y-direction fluctuated drastically, while the isolation coefficient in the Z-direction maintained relatively stable. The isolation coefficients in the Z-direction of the composite subgrade structure were all lower than 0.2, which reflects its great anti-freeze–thaw stability in terms of vibration isolation performance.

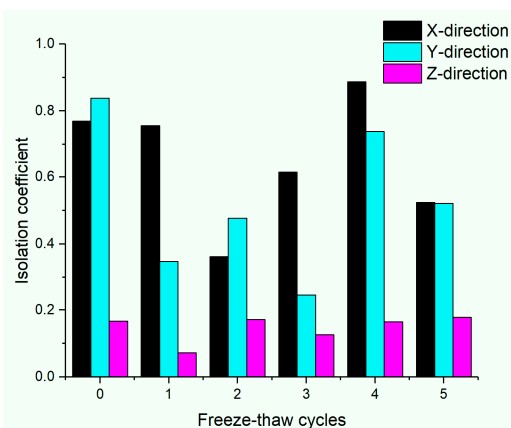

**Figure 10.** The isolation coefficient of the composite subgrade structure after freeze–thaw cycles.

## 4. Conclusions

In this study, XPS plates were adopted to improve the vibration isolation performance of the sustainable subgrade material (silty clay modified by fly ash and crumb rubber). The vibration isolation performance of the composite subgrade structure was evaluated by the indoor impact test. The conclusions can be summarized as follows:

(1) The vibration amplitude of the modified soil subgrade was greater than that of the silty clay. The vibration amplitude of the modified soil subgrade can be significantly reduced by setting XPS plates.

(2) Considering the effect and cost of vibration isolation, 5 cm is considered to be the optimum thickness of XPS plates in road engineering.

(3) XPS plates have excellent vibration isolation stability under different vibration intensities. Laboratory test results showed that the vibration isolation coefficient of the subgrade model with XPS plates in the Z-direction was less than 0.2.

(4) The vibration isolation performance of the composite subgrade structure had outstanding anti-freeze–thaw stability. The vibration isolation coefficient in the Z-direction remained below

0.2 after several freeze–thaw cycles. This shows that the composite subgrade structure can be applied to road engineering in seasonal frozen regions.

In summary, the vibration isolation effect of the composite subgrade structure was proven to be effective. For the modified soil (silty clay modified by fly ash and crumb rubber) subgrade, it will possess more sustainable vibration isolation performance by setting XPS plates. The indoor test results can provide a reference for the design of a composite subgrade structure in practical application.

**Author Contributions:** Conceptualization, L.H. and H.W.; Methodology, H.W.; Software, L.H.; Validation, L.H., H.W. and F.W.; Formal analysis, L.H.; Resources, F.W.; Data curation, L.H.; Writing—original draft preparation, L.H.; Writing—review and editing, L.H. and F.W.; Project administration, H.W.; Funding acquisition, F.W. All authors have read and agreed to the published version of the manuscript.

**Funding:** This research was funded by the National Key R&D Program of China, grant number 2018YFB1600200; Science and Technology Project of Jilin Province Transportation Department, grant number 2017ZDGC6.

**Acknowledgments:** The authors express their appreciation for the financial support of the National Key R&D Program of China (grant number 2018YFB1600200); Science and Technology Project of Jilin Province Transportation Department (grant number 2017ZDGC6). Our thanks go to the editors and reviewers for their efficient work.

**Conflicts of Interest:** The authors declare no conflict of interest.

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
