# Peer review of "Study on the Vibration Isolation Performance of Composite Subgrade Structure in Seasonal Frozen Regions"

_applsci, doi:10.3390/app10103597_

Round 1

Reviewer 1 Report

This paper presented a performance of soil by using XPS plates. It is concluded that the XPS plate significantly improves the vibration behaviour by improving the isolation. The experimental design was systematic and the conclusion was based on the experimental result. However, the paper has a lot of redundancy. There is no need to include figures 1, 2 and 3. These figures should be removed and the text should be changed accordingly.  There is no need for figure 6b which is identical with 6a.  The authors need to clearly label the X and Y coordinate of figure 7. In the experimental section it needs to be clearly mentioned that 3 different XPS plate thickness were used (may be move the table 4 in the experimental section). The paper needs a major restructure to make it easier for the reader to understand. Table 4, 5 and 6 can be combined. Why not marked G1, G2, G3 in Table 4. 

Reviewer 2 Report

1) Fig.4

   It is too small, so required to revise.

2) Fig.7

   It is unclear, so required to revise.

3) Fig.10

  It is unclear, so required to revise. 

Reviewer 3 Report

The reviewed paper presents the results of experimental tests carried out in the laboratory on the road subgrade model. The research aimed to investigate how the damping properties of a road track made of clay modified with fly ash and crumb rubber can be improved by introducing a layer made of XPS plate. On the basis of the results obtained, the Authors recommend the modification of the tested subgrade structure by using a 5cm thick XPS plate that will efficiently improve the damping properties of road foundations. 

The material presented in the thesis is interesting and fairly correctly described, however, after careful reading, the reviewer has the following remarks: 

1)  There is some doubt regarding the representativeness of the results presented. One can see visible differences in measurements in the x and y directions, which, however, have not been commented on by the Authors. It is understandable that the precision of placing the accelerometer in the soil is quite limited, all the more pity that the tests were not repeated, for example, for three analogous containers.

2)  Arbitrary adoption of the dimensions of the test box may also raise considerable doubts. In the analysis of dynamic wave propagation in such a small space, the issue of interference with reflected waves arises. There is no verification to what extent these reflections can be limited by lining the box with XPS plates.

3)  One can doubt whether the approach used by the Authors, in which the dynamic structure response is assessed solely on the basis of the peak acceleration value, is appropriate. It is known that extreme acceleration values strongly depend on the signal sampling, which is why they are quite a dubious measure of the dynamic effect. One can look for some source of inspiration in research procedures used in the analysis of road crash tests.

4)  The description of the conditions under which the tests were carried out lacks information about temperature conditions. Given the significant effect of temperature on the mechanical properties of soil and modified soil components, this should be considered a significant lack. In addition, some tests were carried out after freezing-thawing cycles, which further justifies questions about temperature test conditions. This is even more strange because we can find a temperature sensor in the set of measuring apparatus shown in Figure 6.

5)  Assessing something by “numerical analysis" (see line 302) is, however, something other than numerical processing of experimental data.

6)  There are also some linguistic reservations. To begin with, a general remark regarding the precision of wording in scientific publications, in opposition to colloquial speech. It should be realized that vibrations as such are not measurable. What is measured in the context of vibrations is amplitude, frequency, etc. Therefore, we should not write that vibrations are large or small. 

Detailed language correction notes:

Line 11: “weak” (instead of “wake”)

Line 35: “Costa et al.”

Line 37: “Bajcar et al.”

Line 42: “that is”

Line 48: “many sources” (in place of “many literature”)

Line 56: “durability and thermal” (in place of “durability, thermal”)

Line 56: “Xiang et al.”

Line 58: “Murilo et al. [14]” (instead of “C Murilo [14])

Line 63: “hygroscopicity” (“hydroscopicity” is from another fairy tale)

Line 68: “Ertugrul” (instead of “Ozgur”)

Line 74: “much more progressed” (or “much more developed”)

Line 93: “kPa”

Line 97: “all ingredients were mixed”

Line 103: “Physical”

Line 111 (also Line 115): “sensor DH311E”

Line 111 (also Line 116):  “data acquisition instrument DH5922”

Line 119: “(which is presented in Figure 3) was composed of”

Line 122: It is (“It’s” is informal)  

Line 125: “Figure 4 shows”

Line 128/9: “presented in” (instead of “described as”)

Line 137: “by a hammer”

Line 146: “presented in Figure 6”

Line 154/5: “Accelerations of the fierce vibration could be”

Line 185: “it is”

Line 224: “more significant than”

Line 226: “than for G1”

Line 236: “accords with”

Line 273/4: “increase in the number of”

Line 280/1 (also Line 284): “stability of anti-freeze-thaw resistance”

Line 353: “2005, 41”

Line 355: “Jiao Tong”

Line 358: !!!please notice that position No. 15 is a kind of press announcement or information note without an Author!!! 

Round 2

Reviewer 1 Report

The paper has improved and ready for publication.